# Impact of Body Composition and Sarcopenia on Mortality in Chronic Obstructive Pulmonary Disease Patients

**DOI:** 10.3390/jcm12041321

**Published:** 2023-02-07

**Authors:** Manuel Gómez-Martínez, Wendy Rodríguez-García, Dulce González-Islas, Arturo Orea-Tejeda, Candace Keirns-Davis, Fernanda Salgado-Fernández, Samantha Hernández-López, Angelia Jiménez-Valentín, Alejandra Vanessa Ríos-Pereda, Juan Carlos Márquez-Cordero, Mariana Salvatierra-Escobar, Iris López-Vásquez

**Affiliations:** 1Heart Failure and Respiratory Distress Clinic, Instituto Nacional de Enfermedades Respiratorias “Ismael Cosío Villegas”, Mexico City C.P. 14080, Mexico; 2Licenciatura en Nutriología, Facultad de Estudios Superiores Zaragoza, Universidad Nacional Autónoma de México, Mexico City C.P. 09230, Mexico

**Keywords:** COPD, body composition, strength, muscle mass, mortality, phase angle, impedance ratio

## Abstract

Background: Patients with chronic obstructive pulmonary disease (COPD) have alterations in body composition, such as low cell integrity, body cell mass, and disturbances in water distribution evidenced by higher impedance ratio (IR), low phase angle (PhA), as well as low strength, low muscle mass, and sarcopenia. Body composition alterations are associated with adverse outcomes. However, according to the European Working Group on Sarcopenia in Older People 2 (EWGSOP2), the impact of these alterations on mortality in COPD patients is not well-established. Our aims were to evaluate whether low strength, low muscle mass, and sarcopenia impacted mortality in COPD patients. Methods: A prospective cohort study performance was conducted with COPD patients. Patients with cancer, and asthma were excluded. Body composition was assessed by bioelectrical impedance analysis. Low strength and muscle mass, and sarcopenia were defined according to EWGSOP2. Results: 240 patients were evaluated, of whom 32% had sarcopenia. The mean age was 72.32 ± 8.24 years. The factors associated with lower risk of mortality were handgrip strength (HR:0.91, CI 95%; 0.85 to 0.96, *p* = 0.002), PhA (HR:0.59, CI 95%; 0.37 to 0.94, *p* = 0.026) and exercise tolerance (HR:0.99, CI 95%; 0.992 to 0.999, *p* = 0.021), while PhA below the 50th percentile (HR:3.47, CI 95%; 1.45 to 8.29, *p* = 0.005), low muscle strength (HR:3.49, CI 95%; 1.41 to 8.64, *p* = 0.007) and sarcopenia (HR:2.10, CI 95%; 1.02 to 4.33, *p* = 0.022) were associated with a higher risk of mortality. Conclusion: Low PhA, low muscle strength, and sarcopenia are independently associated with poor prognosis in COPD patients.

## 1. Introduction

Chronic Obstructive Pulmonary Disease (COPD) is a treatable and avoidable illness characterized by persistent respiratory symptoms and progressive and irreversible airflow limitation [1]. COPD is considered a global public health problem because it is the third leading cause of death [2]. Patients with COPD have characteristics such as low cell integrity, body cell mass, disturbances in water distribution between intracellular and extracellular compartments as evidenced by low phase angle (PhA) and higher impedance ratio (IR), as well as low muscle strength, low muscle mass, and sarcopenia [3,4,5,6,7].

Body composition compartments can be assessed by raw bioelectrical impedance analysis (BIA) variables, such as IR and PhA, which provide information on hydration status, cellular mass and quality, nutritional status, and quality of life [8,9,10]. In COPD, some studies have reported significant alterations in these parameters [6,10,11]. In addition, they are independent predictors of mortality [4,6].

According to the updated consensus of the European Working Group on Sarcopenia in Older People 2 (EWGSOP2) published in 2019, sarcopenia is a progressive and generalized skeletal muscle disorder defined as low muscle strength and diminished muscle mass associated with an increased likelihood of adverse outcomes in healthy people and in those with pathologies. In clinical practice for diagnostic sarcopenia, low muscle strength is the primary parameter that defines the role of low muscle mass [12,13,14].

Sarcopenia can occur as a secondary condition in the presence of some diseases such as COPD. In affected patients, multiple factors have been associated with sarcopenia: age, disease severity, disuse, functional performance, oxidative stress, hypoxia, malnutrition, a higher catabolic state, and glucocorticoid use and pro-inflammatory states such as tumor necrosis factor (TNF-α), interleukin (IL)-6, and IL-1β [3,15,16]. The prevalence of sarcopenia in COPD patients is 27.5% (95% CI; 18.4 to 36.5%) across different population settings and other defining categories [17].

Regarding the clinical impact of sarcopenia in COPD patients, several studies have reported lower pulmonary function, functional capacity, quality of life, and higher levels of inflammatory markers [10,11,17,18]. However, the impact of low muscle strength, low muscle mass, and sarcopenia, according to EWGSOP2, on mortality in COPD patients is not well-established. 

In this study, our aims were to evaluate whether low muscle strength, low muscle mass, and sarcopenia, according to EWGSOP2, impacted mortality in COPD patients. 

## 2. Materials and Methods

A prospective cohort study was performed with COPD out-patients at the Instituto Nacional de Enfermedades Respiratorias “Ismael Cosío Villegas” in Mexico City, Mexico, from 30 July 2015 to 31 March 2022.

According to the recommendations of the Global Initiative for Chronic Obstructive Lung Disease (GOLD) [19], patients with a confirmed diagnosis of COPD were included. The subjects were >40 years old, and spirometry with a forced expiratory volume over 1 s (FEV_1_)/forced vital capacity ratio (FVC) ratio < 0.70 post-bronchodilator. Patients with diagnoses of cancer, human immunodeficiency virus, and asthma were excluded. 

### 2.1. Data Collection

Body composition, anthropometry, clinical and demographic variables were evaluated; these are an integral part of the clinical management of patients who come to our Institute. With respect to clinical variables, exacerbations in the previous year were characterized by worsening respiratory symptoms for two or more consecutive days with medication changes and requiring hospital admission. Hospitalization during follow-up was due to acute exacerbation of COPD or acute heart failure.

### 2.2. Anthropometry 

Weight and height were measured according to the manual reference of anthropometric standardization [20]. All subjects wore light clothing and were barefoot. Body mass index was calculated by dividing the total body weight (kilograms) by the squared height (meters). 

### 2.3. Bioelectrical Impedance Analysis (BIA)

Total body composition and raw variables were measured with whole-body BIA using four-pole multifrequency equipment BodyStat QuadScan 4000 (BodyStat, Isle of Man, UK) by standard technique [21]. The BIA method is based on injecting an alternating electric current of a minimal intensity below the sensing thresholds. The impedance Z represents the opposition shown by biological materials to the passage of an alternating electric current. The electrical impedance Z comprises the resistance (R) and reactance (Xc). The current passage determines R through the intracellular and extracellular electrolyte solutions, and the Xc is the delay in current flow measured as a phase shift, reflecting the dielectric properties of the cell mass and integrity of the cell membranes [22].

The measurements were conducted by the same operator, in the morning, in a comfortable area, free of drafts, with portable electric heaters. The area was cleaned before the study. The subjects were fasting and should not have exercised eight hours before or consumed alcohol 12 h before the study. During the entire study, the person was in a supine position with the arms separated from the trunk by about 30° and the legs separated by about 45°. Electrodes were placed on the hand and ipsilateral foot. 

PhA was calculated according to the previous equation: phase angle (degrees) = arctan (Xc/R) · (180/π) [23]. In our population, PhA below the 50th percentile value was <5.1 in men and <4.8 in women.

IR was calculated as follows: the ratio of high (200 kHz) to low frequency (5 kHz) of multifrequency BIA [10]. In our population, IR above the 50th percentile was >0.83 in men and >0.84 in women.

### 2.4. Muscle Mass

Appendicular skeletal muscle mass index (ASMMI) was assessed according to Sergi’s formula [24]: ASMMI (kg/m^2^) = [−3.964 + (0.227 * (Height^2^ (cm)/Resistance) + (0.095 * Weight) + (1.384 * Sex) + (0.064 * Reactance)/Height (m^2^)]. 

Low muscle mass was defined according to EWGSOP2 [14] as ASMMI < 7.0 kg/m^2^, and in women, ASMMI < 6.0 kg/m^2^.

### 2.5. Muscle Strength

Muscle strength was measured with a mechanical Smedley Hand Dynamometer (Stoelting, Wood Dale, UK) according to the technique described in Rodríguez et al. [25].

Low muscle strength was defined according to EWGSOP2 [14] as the presence of low handgrip strength (in men, handgrip strength < 27 kg, and in women, handgrip strength < 16 kg).

### 2.6. Sarcopenia

Sarcopenia was defined according to EWGSOP2 [14] as the presence of low muscle strength (in men, handgrip strength < 27 kg, and in women, handgrip strength < 16 kg) and low muscle mass (in men, ASMMI < 7.0 kg/m^2^, and in women, ASMMI < 6.0 kg/m^2^).

### 2.7. Exercise Tolerance

Exercise tolerance was assessed by a 6-min walk test, which was performed according to American Thoracic Society standards [26].

### 2.8. Pulmonary Function 

Spirometry testing was conducted by an experienced pulmonary technician using a portable spirometer (EasyOnePC, Ndd Medical Technologies Inc., Zürich, Switzerland) according to the criteria of the American Thoracic Society/European Respiratory Society standards [27]. The reference values used for spirometry were obtained in Mexican-American individuals [28].

### 2.9. Endpoint

The endpoint was defined as mortality from all causes.

### 2.10. Statistical Analysis

Analyses were performed using a commercially available STATA version 14 (Stata Corp., College Station, TX, USA). Categorical variables were presented as frequencies and percentages. The Shapiro–Wilk test was used to test the normality of continuous variables; continuous variables with normal distribution were presented as mean and standard deviation, and non-normal variables were presented as median and percentiles 25–75. A comparison among study groups was analyzed with a chi-square test or Fisher’s F test for categorical variables and unpaired Student’s *t*-test or Mann–Whitney U tests for continuous variables. Finally, bivariate Cox’s proportional hazards analysis was performed to evaluate the impact of body composition and sarcopenia on mortality. Subsequently, the multivariate Cox proportional hazards model was adjusted by clinical variables *p* < 0.100 in the bivariate model. Due to collinearity, body composition and BODE index variables were not included in the model. *p* < 0.05 was considered statistically significant.

## 3. Results

Two hundred forty patients with COPD were evaluated. During 6.66 years of follow-up, 31 subjects died. The median survival time was 941 days (range 411–1343.5). The mean population age was 72.32 ± 8.24 years; 51.25% were men; 25.83% had diabetes, 49.17% had systemic hypertension, 69.58% were overweight or obese, 42.08% were in heart failure, and 32% had sarcopenia; 22.5% had hospitalization during the follow-up, and 17.92% had exacerbations in the previous year. The mean of the 6-min walk test distance was 319.04 m ± 133.06 (Table 1). 

Patients who succumbed were older and had a higher BODE index, more hospitalizations during the follow-up and more exacerbations in the previous year, as well as lower PhA. They had a higher prevalence of low muscle strength, sarcopenia, and less exercise tolerance than patients who survived (Table 1). 

In the bivariate model, age, BODE index, PhA below the 50th percentile, low muscle strength, and sarcopenia were associated with more mortality risk. In contrast, PhA was associated with less mortality risk (Table 2). 

Multivariate analysis showed that handgrip strength (HR: 0.91, CI 95%; 0.85 to 0.96, *p* = 0.002), PhA (HR: 0.59, CI 95%; 0.37 to 0.94, *p* = 0.026), and exercise tolerance (HR: 0.99, CI 95%; 0.992 to 0.999, *p* = 0.022) were associated with a lower risk of mortality, while PhA below the 50th percentile (HR: 3.47, CI 95%; 1.45 to 8.29, *p* = 0.005), low muscle strength (HR: 3.49, CI 95%; 1.41 to 8.64, *p* = 0.007) and sarcopenia (HR: 2.10, CI 95%; 1.02 to 4.33, *p* = 0.043) were associated with a higher risk of mortality (Table 3 and Figure 1).

## 4. Discussion

The main finding was the impact of body composition components such as handgrip strength, PhA, low muscle strength, and sarcopenia on prognosis in COPD patients. 

COPD patients have significant body composition alterations, including sarcopenia. In our study, the prevalence was 32%, independently associated with a two-fold increase in the risk of death (HR: 2.10, CI 95%; 1.02 to 4.33, *p* = 0.043). Similar results were found by Schols et al., who showed that COPD patients with cachexia, evaluated by low fat-free mass index (FFMI) and low BMI, have a greater mortality risk than patients without impaired FFMI [29]. Benz et al. showed that sarcopenia is an independent risk factor for mortality in COPD patients [30]. Similar results have been found in diverse populations, such as geriatric [13] and dialysis patients [31]. In addition, sarcopenia is associated with functional disability, a higher rate of falls, a risk of hospitalization incidents, and lower pulmonary function [11,12]. 

Muscle strength is presently the most reliable measure of muscle function; low muscle strength is the primary parameter for diagnosing sarcopenia, according to EWGSOP2, and it is defined as probable sarcopenia [14]. In addition, decreased handgrip strength is significantly associated with moderate-to-very severe airflow limitation in the general population [32]. Our results showed that subjects who did not survive had lower handgrip strength than patients who survived. Subjects with low muscle strength had a threefold increase in the risk of death adjusted for confounding variables. Similar results have been observed in COPD and other populations [5,33]. 

In a healthy aging population, muscular strength declines 1–2% per year [32], while for COPD patients this decline is estimated at 4.3% per year [34]. The pathological mechanisms involved in this muscle dysfunction in COPD subjects are corticosteroid use, hypoxemia, systemic inflammation, and oxidative stress. These factors play a fundamental role in affecting anabolism, and are associated with protein catabolism, mitochondrial dysfunction, lower plasma amino acids, and reduction of type 1 fibers in the peripheral muscles, among others [35]. Byun et al. showed a negative correlation between handgrip strength and skeletal muscle mass index with IL-6 and TNF-α but this was not found with skeletal muscle mass index. Higher TNF-α levels were also associated with sarcopenia in patients with stable COPD [36]. 

Skeletal muscle is the human body’s largest organ and accounts for 40–50% of total body weight under physiological conditions. In COPD patients with low FFMI, lower muscle strength has been observed compared with normal FFMI [4]. Regarding the impact of low muscle mass on prognosis, the evidence is not conclusive; some studies show that low muscle mass is a predictor of overall mortality in some populations, such as COPD patients and geriatric people [37,38]. In contrast, other studies have observed no association with exacerbations, days of hospitalization, quality of life, and mortality [6,18]. In our study, no association was observed between mortality and ASMMI or low muscle mass defined according to EWGSOP2 (in men, ASMMI < 7.0 kg/m^2^ and in women, ASMMI < 6.0 kg/m^2^) [5]. Although the impact of muscle mass is controversial, skeletal muscle is an endocrine organ with multiple metabolic functions such as energy homeostasis, heat regulation, insulin sensitivity, and amino acid metabolism [39].

Another of the determining factors in the prognosis of COPD patients is exercise tolerance. In our study, we observed that per meter increase in the 6-min walk test, there is a 1% reduction in the probability of mortality (HR: 0.99, CI 95%; 0.992 to 0.999); similar results have been observed in different studies [6,40]. 

Low BMI is a significant risk factor for mortality in COPD [41,42] This implies that the subject has a cachexia diagnosis [43]. However, BMI has severe limitations: it is calculated as the ratio of body weight to height squared, which does not allow determining the distribution between the different components of body composition, such as muscle mass, fat mass, and hydration status, which have an independent impact on the prognosis of the subjects [6,29]. In our study, BMI was not associated with a worse prognosis, probably due to the high prevalence of overweight or obesity. A high prevalence of low muscle mass and muscle strength were observed. When determining the impact of these variables on the prognosis of patients with COPD, we observed that muscle strength is a stronger predictor for mortality than BMI or muscle mass. Similar results have been observed in other studies [6,29].

PhA and IR are raw BIA variables that provide information on water distribution between intracellular and extracellular compartments and cellular integrity [6,8,9].

Concerning PhA, in this study, non-surviving patients had lower PhA than patients who survived, and PhA below the 50th percentile was independently associated with mortality. Similar results were observed by Maddocks et al., who showed that COPD patients with low PhA (<4.5°) had lower quadriceps strength and quality of life, as well as more exacerbations and hospital admissions [4]. Moreover, low PhA is associated with an impaired pulmonary function [44]. 

Likewise, COPD patients have been observed to be malnourished and malnourished with systemic inflammation, and sarcopenic subjects have higher values of IR. Other studies on COPD patients showed that subjects with IR > 0.84 had decreased handgrip strength, skeletal muscle mass index, and diffusion capacity for carbon monoxide than subjects with IR < 0.84. In addition, IR > 0.84 was associated with lower FEV_1_ and FVC [44]. Blasio et al. showed that IR is an independent predictor of all-cause mortality in COPD [6]. However, in our study, IR above the 50th percentile was not associated with mortality in the bivariate and multivariate model.

It is vital to evaluate and identify alterations in body composition, such as low PhA, low muscle strength and sarcopenia, since early attention to these alterations can positively impact clinical variables and the prognosis of patients with COPD. In a meta-analysis of randomized controlled clinical trials, Bernardes et al. evaluated the effect of energy and/or protein oral nutritional supplements or food fortification on body composition parameters. They found improvement in midarm muscle circumference, handgrip strength, and lean body mass [44]. In addition, pulmonary rehabilitation also showed improvement in exercise tolerance, quality of life, respiratory muscle strength, etc. [45,46].

A low FEV1 has been associated with an increased risk of hospitalizations and death in COPD patients [41,47]. However, in this study, no differences were observed between the survivor and non-survivor groups. These may be because other risk factors impact the prognosis of COPD subjects, such as short-distance walking, low body mass index, and a high degree of functional breathlessness. The BODE index is a multidimensional index of disease severity in COPD that incorporates these risk factors, which is a better predictor of prognosis in COPD patients [41,48]. In our study, the BODE index was associated with a worse prognosis.

### Strengths and Limitations of the Study

Among the limitations of this study is the moderate sample size and a long recruitment period, which could impact the representativeness of the data. However, our study’s strength is that it is a prospective cohort, so we were able to assess the causality, and we also performed multivariate models, which allowed us to adjust for possible confounding variables.

## 5. Conclusions

Low PhA, low muscle strength, and sarcopenia are independently associated with poor prognosis in COPD patients.

## Figures and Tables

**Figure 1 jcm-12-01321-f001:**
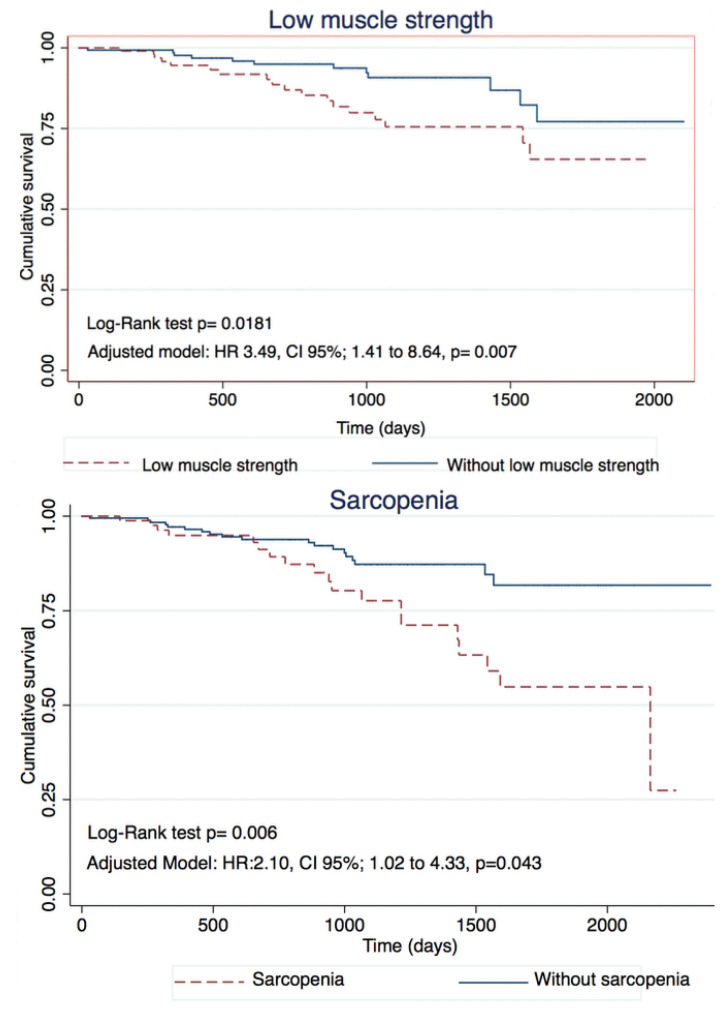
Kaplan–Meier curves for all cause death.

**Table 1 jcm-12-01321-t001:** Clinical and body composition characteristic in surviving and non-surviving subjects.

	Total Population *n* = 240	Non Survival *n* = 31	Survival *n* = 209	*p*-Value
**Clinical variables**				
Age, years	72.32 ± 8.24	78.70 ± 7.16	71.37 ± 8.66	<0.001
Male, *n* (%)	123 (51.25)	19 (61.29)	104 (49.76)	0.231
Diabetes, *n* (%)	62 (25.83)	4 (12.90)	58 (27.75)	0.078
Hypertension, *n* (%)	118 (49.17)	16 (51.61)	102 (48.80)	0.770
Overweight or obesity, *n* (%)	167 (69.58)	17 (54.84)	150 (71.77)	0.056
Heart Failure, *n* (%)	101 (42.08)	17 (54.84)	84 (40.19)	0.123
FEV_1_, (%)	57.61 ± 23.55	62.25 ± 23.57	56.94 ± 23.54	0.283
FEV_1,_ (L)	1.26 ± 0.63	1.23 ± 0.57	1.27 ± 0.64	0.772
FEV_1_/FVC	0.55 ± 0.15	0.59 ± 0.17	0.55 ± 0.15	0.184
**GOLD Stage, *n* (%)**				
1–2	133 (60.18)	19 (67.86)	114 (59.07)	
3–4	88 (39.82)	9 (32.14)	79 (40.93)	0.375
BODE index	3 [2 to 4]	4 [2 to 5]	3 [1 to 4]	<0.001
Hospitalization, *n* (%)	54 (22.50)	13 (41.94)	41 (19.62)	0.005
Length of hospital stay, d	8 [3,4,5,6,7,8,9,10,11,12,13,14]	8 [1,2,3,4,5,6,7,8,9,10,11,12,13,14,15,16,17]	8 [4,5,6,7,8,9,10,11,12]	0.793
Exacerbations in the previous year, *n* (%)	43 (17.92)	10 (32.26)	33 (15.79)	0.026
**Body composition**				
Height, cm	157.02 ± 11.76	156.28 ± 11.97	157.12 ± 11.76	0.709
Weight, kg	70.02 ± 18.35	65.60 ± 14.61	70.67 ± 18.78	0.151
Body Mass Index, kg/m^2^	28.44 ± 7.43	26.67 ± 5.53	28.70 ± 7.65	0.156
Fat Mass, kg	39.83 ± 10.13	38.56 ± 7.80	39.97 ± 10.35	0.611
Fat Free Mass Index, kg/m	16.62 ± 3.42	16.57 ± 2.38	16.63 ± 3.54	0.955
Total body water, %	51.64 ± 8.17	53.18 ± 7.05	51.40 ± 8.32	0.274
Extracellular water, %	27.77 ± 6.03	30.01 ± 3.84	27.48 ± 6.22	0.155
Impedance ratio	0.83 [0.81–0.85]	0.84 [0.83–0.86]	0.83 [0.81–0.85]	0.149
Handgrip strength, kg	22.75 ± 8.63	20.68 ± 8.77	23.02 ± 8.60	0.213
Phase angle, °	5.05 ± 0.96	4.62 ± 1.06	5.11 ± 0.95	0.008
ASMMI, kg/m^2^	6.85 ± 1.17	6.70 ± 0.90	6.88 ± 1.20	0.336
Low muscle strength, *n* (%)	91 (44.61)	16 (66.67)	75 (41.67)	0.021
Low muscle mass, *n* (%)	93 (39.57)	16 (51.61)	77 (37.75)	0.141
Sarcopenia, *n* (%)	77 (32.08)	17 (54.84)	60 (28.71)	0.004
Exercise tolerance, m	319.04 ± 133.06	259.55 ± 175.50	327.87 ± 123.68	0.043

FEV_1,_ Forced Expiratory Volume in the first second; FEV_1_/FVC, Forced expiratory volume over 1 s/forced vital capacity ratio; IR, Impedance ratio; GOLD stage, Global Initiative for Chronic Obstructive Lung Disease stage; ASMMI, Appendicular Skeletal Muscle Mass Index.

**Table 2 jcm-12-01321-t002:** Bivariate analysis of body composition predictors of mortality in COPD patients.

	HR	CI 95%	*p*-Value
Age, years	1.11	1.05 to 1.17	0.001
Male	1.95	0.94 to 4.03	0.069
Diabetes	0.48	0.14 to 1.50	0.089
Hypertension	1.12	0.55 to 2.28	0.739
Overweight or obesity	0.59	0.29 to1.21	0.157
Heart Failure	1.92	0.94 to 3.93	0.072
FEV_1_, (%)	1.00	0.98 to 1.02	0.651
FEV_1,_ (L)	0.91	0.47 to 1.79	0.805
FEV_1_/FVC	3.45	0.29 to 40.73	0.325
III-IV GOLD Stage	1.05	0.68 to 1.62	0.793
BODE index	1.50	1.24 to 1.83	<0.001
Hospitalization	1.77	0.86 to 3.65	0.119
Length of hospital stay	1.06	0.97 to 1.03	0.654
Exacerbations in the previous year	1.91	0.89 to 4.06	0.093
**Body composition**			
Height, cm	1.00	0.97 to 1.03	0.764
Weight, kg	0.98	0.96 to 1.01	0.325
Body Mass Index, kg/m^2^	0.96	0.90 to 1.01	0.185
Fat Mass, %	0.97	0.92 to 1.02	0.334
Fat Free Mass Index, kg/m	0.98	0.83 to 1.16	0.844
Total body water, %	1.02	0.97 to 1.06	0.351
Extracellular water, %	1.01	0.96 to 1.06	0.577
IR above 50th percentile	1.87	0.88 to 3.96	0.102
Handgrip strength, kg	0.97	0.92 to 1.02	0.275
Phase angle, °	0.54	0.35 to 0.84	0.007
PhA below 50th percentile	4.04	1.73 to 9.41	0.001
ASMMI, kg/m^2^	0.96	0.70 to 1.33	0.846
Low muscle strength	2.76	1.17 to 6.46	0.019
Low muscle mass	1.64	0.80 to 3.35	0.176
Sarcopenia	2.57	1.26 to 5.25	0.009
Exercise tolerance, per meter increase	0.99	0.992 to 1.00	0.061

FEV_1_, Forced expiratory volume over 1 s; FEV_1_/FVC, Forced expiratory volume over 1 s/forced vital capacity ratio; IR, Impedance ratio; PhA, Phase angle; ASMMI, Appendicular Skeletal Muscle Mass Index.

**Table 3 jcm-12-01321-t003:** Multivariate analysis of body composition predictors of mortality in COPD patients.

	HR	CI 95%	*p*-Value
Height, cm	1.00	0.97 to 1.03	0.764
Weight, kg	0.98	0.96 to 1.01	0.325
Body Mass Index, kg/m^2^	0.98	0.96 to 1.01	0.325
Fat Mass, %	1.05	0.94 to 1.17	0.367
Fat Free Mass Index, kg/m	0.94	0.77 to 1.15	0.878
Total body water, %	1.00	0.95 to 1.06	0.730
Extracellular water, %	1.01	0.95 to 1.06	0.683
IR above 50th percentile	1.61	0.73 to 3.52	0.231
Handgrip strength, kg	0.91	0.85 to 0.96	0.002
Phase angle, °	0.59	0.37 to 0.94	0.026
PhA below 50th percentile	3.47	1.45 to 8.29	0.005
ASMMI, kg/m^2^	0.88	0.59 to 1.31	0.539
Low muscle strength	3.49	1.41 to 8.64	0.007
Low muscle mass	1.49	0.72 to 3.10	0.277
Sarcopenia	2.10	1.02 to 4.33	0.043
Exercise tolerance, per meter increase	0.99	0.992 to 0.999	0.022

IR, Impedance ratio; PhA, Phase angle; ASMMI, Appendicular Skeletal Muscle Mass Index.

## Data Availability

Not applicable.

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
