# Peer review of "Impact of Body Composition and Sarcopenia on Mortality in Chronic Obstructive Pulmonary Disease Patients"

_jcm, 2023, doi:10.3390/jcm12041321_

Round 1

Reviewer 1 Report

A prospective study conducted on a small cohort of patients with COPD in order to highlight the correlates of any cause mortality. In general, the greater prognostic accuracy of a multivariate indicator is confirmed, when  compared with the predictive capabilities of a single indicator, as could be a measure of impaired respiratory function in these patients.In particular, the very significant prognostic role of sarcopenia and exercise capacity is highlighted. I have a couple of major questions. in this cohort, FEV1 and FEV/FVC were a bit higher in those who died than in survivors and, of course, they didn't predict mortality.  This result is contradictory with some multivariate well known predictors, as BODE index; still referred to also in the 2022 GOLD document. We all recognize the partial predictive role of FEV1 with respect to all-cause mortality in COPD patients, but denying any predictive ability of FEV1 forces the authors to discuss this point in great detail. Also the BMI was quite similar between died and survivors.This result denies any prognostic role of BMI and this point should also be discussed in detail because BMI is included in the multivariate prognostic indices GOLD refers to. Finally, I think that it should be interesting for the readers, to know why an index of dyspnea, such as the mMRC index, and some clinical characteristics, such as the history of exacerbations, were not taken into account as predictors of mortality in COPD.  

Author Response

Impact of body composition and sarcopenia on mortality in Chronic Obstructive Pulmonary Disease patients

Manuscript ID: jcm-2116758

Reviewer 1

A prospective study conducted on a small cohort of patients with COPD in order to highlight the correlates of any cause mortality. In general, the greater prognostic accuracy of a multivariate indicator is confirmed, when  compared with the predictive capabilities of a single indicator, as could be a measure of impaired respiratory function in these patients. In particular, the very significant prognostic role of sarcopenia and exercise capacity is highlighted. I have a couple of major questions. in this cohort, FEV1 and FEV/FVC were a bit higher in those who died than in survivors and, of course, they didn't predict mortality. This result is contradictory with some multivariate well known predictors, as BODE index; still referred to also in the 2022 GOLD document. We all recognize the partial predictive role of FEV1 with respect to all-cause mortality in COPD patients, but denying any predictive ability of FEV1 forces the authors to discuss this point in great detail.

The authors are grateful for the comments on our work; we believe they will be instrumental in improving its quality.

The authors agree with the comment, so we added the BODE index variable in the results section and discussed this point.

“A low FEV1 has been associated with an increased risk of hospitalizations and death in COPD patients [36,42]. In this study, no differences were observed between the survivor and non-survivor groups. These may be because other risk factors impact the prognosis of COPD subjects, such as short-distance walking, low body mass index, and a high degree of functional breathlessness. However, the BODE index is a multidimensional index of disease severity in COPD that incorporates these risk factors, and has been reported as a better predictor of prognosis in COPD patients [36,43]. In our study, the BODE index was associated with a worse prognosis.”

Also the BMI was quite similar between died and survivors.This result denies any prognostic role of BMI and this point should also be discussed in detail because BMI is included in the multivariate prognostic indices GOLD refers to.

Low BMI is a significant risk factor for mortality in COPD [36,37]BMI < 20 implies a cachexia diagnosis [38]. However, BMI has severe limitations it is calculated as the ratio of body weight to height squared, which does not allow discriminating between the different components of body composition, such as muscle mass, fat mass, and hydric status, which have an independent impact on the prognosis of the subjects [6,25]. In our study, BMI was not associated with a worse prognosis, probably due to the high prevalence of overweight or obesity. Besides, a high prevalence of low muscle mass and muscle strength were observed. When determining the impact of these variables on the prognosis of patients with COPD, we observed that muscle strength is a stronger predictor for mortality than BMI and muscle mass. Similar results have been observed in other studies [6,25].”

Finally, I think that it should be interesting for the readers, to know why an index of dyspnea, such as the mMRC index, and some clinical characteristics, such as the history of exacerbations, were not taken into account as predictors of mortality in COPD.  

The authors agree with the comments and add the BODE index, hospitalizations, and previous exacerbations variables in the results section  to better describe the population.

“Patients who succumbed were older and had more BODE index, hospitalizations, previous exacerbations previous exacerbations...”

“In the bivariate model, age, BODE index, PhA below the 50th percentile, low muscle strength, and sarcopenia were associated with more mortality risk... ”

The authors appreciate the comments and would be grateful if you could let us know if any new observations could improve the quality of the manuscript.

Reviewer 2 Report

This study aimed to evaluate whether low muscle strength, low muscle mass, and sarcopenia, according to EWGSOP2, impacted mortality in COPD patients. Finally, concluded that PhA, low muscle strength, and sarcopenia are independently associated with poor prognosis in COPD patients.
1. However, this study is not innovative. Previous studies had confirmed that PhA (PMID: 30659818) and sarcopenia (PMID: 31133471, PMID: 35036418) are independently associated with mortality in COPD patients according to EWGSOP2.
2. In addition, as described in Methods and Results “the COPD included from August 1st, 2013, to March 31, 2022...” and “During the median length of follow-up of 941 days (range 411 - 1343.5)...”. I can't understand the designs of follow-up time. During ten years of study, the sample size was only 240?So, the results are not representative.
3. The author emphasized that  this study is a prospective cohort, and able to assess the causality, as well as performed multivariate models. In fact, I did not confirm that whether the authors took into account collinearity between variables. For example, sarcopenia was defined as the presence of low muscle strength (in men, handgrip strength < 27 kg, and in women, handgrip strength < 16 kg) and low muscle mass (in men, ASMMI < 7.0 kg/m2, and in women, ASMMI <6.0 kg/m2). However, the multivariate model included the variables of sarcopenia, low muscle strength and low muscle mass. I didn't find the results of the collinearity test. Therefore, I don't think the results are credible.
4. As described in results, exercise tolerance were associated with less mortality risk, but the p-value was 0.061.
5. PhA  (HR: 0.61, CI95%; 0.38 to 0.96, p= 0.035) is independently associated with poor prognosis in COPD patients?? please check the results carefully.

Author Response

Impact of body composition and sarcopenia on mortality in Chronic Obstructive Pulmonary Disease patients

Manuscript ID: jcm-2116758

Reviewer 2

This study aimed to evaluate whether low muscle strength, low muscle mass, and sarcopenia, according to EWGSOP2, impacted mortality in COPD patients. Finally, concluded that PhA, low muscle strength, and sarcopenia are independently associated with poor prognosis in COPD patients.

  1. However, this study is not innovative. Previous studies had confirmed that PhA (PMID: 30659818) and sarcopenia (PMID: 31133471, PMID: 35036418) are independently associated with mortality in COPD patients according to EWGSOP2.

The authors are grateful for the comments on our work; we believe they will be instrumental in improving its quality.

Undoubtedly, the authors mentioned above have contributed significantly to the knowledge about alterations in body composition and their impact on patient prognosis. However, the authors of this manuscript believe that our work contributes to the knowledge since the populations have different characteristics, such as the prevalence of comorbidities and differences in body composition due to ethnicity. In addition, differences in the instruments used for the diagnosis of sarcopenia.

The authors have added the Benz et al., study to the discussion section.

“Besides, Benz et al., showed that sarcopenia is an independent risk factor for mortality in COPD patients [26] ”

2. In addition, as described in Methods and Results “the COPD included from August 1st, 2013, to March 31, 2022...” and “During the median length of follow-up of 941 days (range 411 - 1343.5)...”. I can't understand the designs of follow-up time. During ten years of study, the sample size was only 240?So, the results are not representative.

The authors thank you for the comment, and we note a severe drafting mistake in the methods section, as the study started on July 30, 2015. Thus, the follow-up time was 6.66 years. However, we agree that six years of follow-up is a long period for the recruitment of 240 patients, and this could contribute to significant bias.

The authors corrected the study start date in the methods section and added the study follow-up and mean survival time. Also, in the limitations section, we declared the possible bias caused by the long follow-up period.

"A prospective cohort study performance in COPD out-patients at the Instituto Nacional de Enfermedades Respiratorias "Ismael Cosío Villegas" in Mexico City, Mexico, from July 30, 2015, to March 31, 2022."

"Two hundred forty patients with COPD were evaluated. During 6.66 years of length follow-up, 31 subjects died. The median survival time was 941 days (range 411 - 1343.5). "

"Among the limitations of this study is the moderate sample size and a long recruitment period, which could impact the representativeness of the data. “

3. The author emphasized that  this study is a prospective cohort, and able to assess the causality, as well as performed multivariate models. In fact, I did not confirm that whether the authors took into account collinearity between variables. For example, sarcopenia was defined as the presence of low muscle strength (in men, handgrip strength < 27 kg, and in women, handgrip strength < 16 kg) and low muscle mass (in men, ASMMI < 7.0 kg/m2, and in women, ASMMI <6.0 kg/m2). However, the multivariate model included the variables of sarcopenia, low muscle strength and low muscle mass. I didn't find the results of the collinearity test. Therefore, I don't think the results are credible.

The authors agree with the comment and appreciate the observation. Collinearity can lead to unstable covariate estimates and mask or amplify a covariate's effect. Indeed, collinearity exists between several body composition variables, such as sarcopenia with hand strength and muscle mass; phase angle and impedance index; BMI with height, weight, body fat, or skeletal muscle mass.

That is why,  in Table 3, body composition variables were adjusted for clinical variables such as age, sex, diabetes, heart failure, and previous exacerbations. However, the wording of the manuscript is not precise on this point. Therefore, we modified the wording of the statistical analysis.

“Subsequently, the multivariate Cox proportional hazards model was adjusted by clinical variables p< 0.100 in the bivariate model. Due to collinearity, body composition and BODE index variables were not included.”

4. As described in results, exercise tolerance were associated with less mortality risk, but the p-value was 0.061.

The authors agree with the observation, and we modified the redaction.

“...In contrast, PhA was associated with less mortality risk (Table 2).”

5. PhA  (HR: 0.61, CI95%; 0.38 to 0.96, p= 0.035) is independently associated with poor prognosis in COPD patients?? please check the results carefully.

The author agree with the comment and modified the redaction.

“Low PhA, low muscle strength, and sarcopenia are independently associated with poor prognosis in COPD patients”

The authors appreciate the comments and would be grateful if you could let us know if any new observations could improve the quality of the manuscript.

Reviewer 3 Report

Dear authors,

I liked your manuscript.

I have only a few suggestions:

Point 1. It is necessary to provide a reference for the claims on p 3 lines 99 and 102

Point 2. It would be significant to know what the survival time was, you only listed the length of follow-up

Point 3. Unfortunately, there is no data on exacerbations and hospital admissions for survivors and non-survivors, which should be mentioned as a study limitation.

Point 4. Please align the references with the style of the journal

Author Response

Impact of body composition and sarcopenia on mortality in Chronic Obstructive Pulmonary Disease patients

Manuscript ID: jcm-2116758

Reviewer 3

Dear authors,

I liked your manuscript.

 I have only a few suggestions:

 The authors are grateful for the comments on our work.

Point 1. It is necessary to provide a reference for the claims on p 3 lines 99 and 102

The authors agree with the comment and add a reference to lines 99 and 102

“PhA was calculated according to the previous formula [22]. In our population, PhA below 50th percentile value was < 5.1 in men and < 4.8 in women.

IR was calculated as follows: the ratio of high (200 kHz) to low frequency (5 kHz) of multifrequency BIA [10]. In our population, IR above the 50th percentile was > 0.83 in men and > 0.84 in women.”

Point 2. It would be significant to know what the survival time was, you only listed the length of follow-up.

The authors agree with the comment and describe the time of follow-up and the survival time.

“During 6.66 years of length follow-up, 31 subjects died. The median of survival time was 941 days (range 411 - 1343.5)”

Point 3. Unfortunately, there is no data on exacerbations and hospital admissions for survivors and non-survivors, which should be mentioned as a study limitation.

The authors agree with the comment. Fortunately, with hospital admissions, and previous exacerbations data, we add this data to the results section and tables. And the multivariate models were adjusted by “hospital admissions.”

“Patients who succumbed were older and had more BODE index, hospitalizations, previous exacerbations previous exacerbations ...”

Point 4. Please align the references with the style of the journal

We align the references in the style of the journal.

The authors appreciate the comments and would be grateful if you could let us know if any new observations could improve the quality of the manuscript.

Round 2

Reviewer 1 Report

Thank you for your answers to my observations and for the corrections in the manuscript.

Author Response

Thank you for taking the time to review the article. The authors are grateful for your feedback. 
